# Efficacy of the Combined Intrastromal Injection of Voriconazole and Amphotericin B in Recalcitrant Fungal Keratitis

**DOI:** 10.3390/microorganisms12050922

**Published:** 2024-04-30

**Authors:** Antonio Moramarco, Arianna Grendele, Danilo Iannetta, Simone Ottoboni, Giulia Gregori, Natalie di Geronimo, Margherita Ortalli, Tiziana Lazzarotto, Luigi Fontana

**Affiliations:** 1Ophthalmology Unit, Dipartimento di Scienze Mediche e Chirurgiche, Alma Mater Studiorum, University of Bologna, 40126 Bologna, Italy; grendele.arianna@gmail.com (A.G.); simone.ottoboni@studio.unibo.it (S.O.); natalie.digeronimo2@unibo.it (N.d.G.); luifonta@gmail.com (L.F.); 2IRCCS Azienda Ospedaliero Universitaria di Bologna, 40138 Bologna, Italy; 3Department of Organs of Sense, University of Rome La Sapienza, 00185 Rome, Italy; iannettadanilo@gmail.com; 4Eye Clinic, Department of Experimental and Clinical Medicine, Polytechnic University of Marche, 60131 Ancona, Italy; g.gregori98@gmail.com; 5Department of Medical and Surgical Sciences, Alma Mater Studiorum, University of Bologna, 40138 Bologna, Italy; margherita.ortalli2@unibo.it (M.O.); tiziana.lazzarotto@unibo.it (T.L.); 6Microbiology Unit, IRCCS Azienda Ospedaliero Universitaria di Bologna, 40138 Bologna, Italy

**Keywords:** infection, ocular surface, fungal keratitis, antifungal intrastromal injection, therapeutic penetrating keratoplasty, ophthalmology

## Abstract

This study aims to report the efficacy of a combined intrastromal injection in optimizing the outcome of severe mycotic keratitis. Herein, we report a case series of 20 consecutive patients with positive fungal cultures not responding to topical antifungal treatment. Patients received cycles of intrastromal injections of voriconazole (50 µg/0.1 mL) and amphotericin B (2.5 µg/0.1 mL); all patients continued their topical antifungal therapy. The organisms isolated were *Fusarium* (*n* = 5), *Aspergillus* (*n* = 4), *Candida* (*n* = 4), *Rhodotorula* (*n* = 2), *Penicillium* (*n* = 2), *Alternaria* (*n* = 1), *Bipolaris* (*n* = 1), and *Curvularia* (*n* = 1). The size of the infiltrate varied from 6.5 to 1.5 mm. At presentation, the best corrected visual acuity (BCVA, namely, the best visual acuity achieved with glasses, if needed) was less than 20/400 in all patients, improving to better than 20/400 in eleven patients. Seven patients required surgical intervention; four of them underwent penetrating keratoplasty (PK) à chaud one month after the first intrastromal injection. Patients who underwent surgery achieved a BCVA of 20/40 or better. Combined intrastromal injections before therapeutic penetrating keratoplasty (TPK) effectively reduced ulcer size and graft diameter, preventing infection recurrence. Our results highlight the efficacy of combined intrastromal injections in optimizing outcomes for severe mycotic keratitis undergoing TPK.

## 1. Introduction

Fungal keratitis is a major cause of blindness, especially in tropical and subtropical countries [1], where 20–60% of the corneal infections with positive culture are of fungal etiology [2]. Between 1.5 and 2 million new cases of blindness associated with keratitis are reported each year in developing countries [3]. The global incidence of fungal keratitis each year is 1.4 million. Approximately 10% of the eyes perforate and 60% of the patients are left with monocular blindness [2]. Managing this disease is challenging due to its insidious clinical presentation and poor response to standard antifungal treatment. Mycotic infection must be considered in case of leading risk factors, such as corneal trauma caused by plants, indiscriminate instillation of topical corticosteroids, or prolonged contact lens use [4].

Treatment for fungal keratitis is usually carried out with topical antifungal agents. Topical natamycin is usually a first-line treatment [5]; although, alternative topical therapies can be considered, including voriconazole, chlorhexidine, amphotericin B, and econazole. However, most topical antifungal drops have low penetration into the deeper layers of the cornea, which leads to an unsatisfactory therapeutic concentration at the infection site [6], a suboptimal therapeutic response, prolonged therapy, and poor outcomes [7]. Therefore, a surgical approach, such as therapeutic penetrating keratoplasty (TPK), is usually carried out in fungal infections more frequently than bacterial keratitis [8]. Furthermore, TPK has a higher graft failure rate than that performed for bacterial etiology [9]. Thus, a different approach is mandatory to improve corneal transplant outcomes in recalcitrant cases.

Targeted drug delivery, such as an intrastromal injection of voriconazole and amphotericin B, has shown promising results in achieving an adequate drug concentration at the injection site [10,11], potentially improving the management of severe fungal keratitis.

In this case series, we report the outcomes of the combined intrastromal injection of voriconazole and amphotericin B in twenty patients with a deep mycotic infection unresponsive to topical antifungal agents.

## 2. Materials and Methods

This study was conducted in accordance with the Declaration of Helsinki and the CE AVEC: 901/2022/Oss/AOUBo. Informed consent was obtained from all subjects. Twenty eyes of twenty consecutive patients with a positive culture for fungal agents poorly responding to topical natamycin 5% concentration (50 mg/mL) and voriconazole 1% concentration (10 mg/mL) were prospectively included from June 2020 to June 2023. We define a poor response as the absence of variation in the size and depth of the ulcer or infiltrate during the 7–10 days of ocular topical treatment with natamycin 5% and voriconazole 1%. Exclusion criteria were corneal perforation or impending perforation, mixed keratitis, sclera involvement, concomitant endophthalmitis, and suspected autoimmune conditions.

The definitive diagnosis of fungal infection was based on positive cultures carried out on corneal tissue obtained by scraping and clinical assessment. Corneal scrapings were obtained under topical anesthesia in a standardized manner and sent for microbiological investigations, such as Gram-stained smears and seeding on blood and chocolate agar. Targeted antifungal therapy was started as soon as fungal etiology was suspected but only after corneal scraping was performed.

At first presentation, for each patient, we collected clinical history and performed a clinical assessment, which included slit lamp biomicroscopy with evaluation of the size and depth of the corneal infiltrate, epithelial defect extent, presence and height of hypopyon, and cest corrected visual acuity (BCVA). Anterior segment photographs were recorded at each visit.

Keratitis was defined as “worsened” if there was a noted increase of 20% in the size and depth of the infiltrate while it was defined as “healing” if the ulcer and the infiltrate were reduced by more than 20%. The infection was considered non-responsive to topical therapy if there was no change or a worsening at the slit lamp examination [7].

Recurrence after TPK was assessed based on signs of infections at the slit lamp examination, such as the recurrence of stromal infiltrate at the host side extending to the donor, endothelial exudates, and hypopyons. Where in doubt, the diagnosis was confirmed by a new positive scraping for the same organism of the primary infection [12].

Finally, we considered signs of toxicity of the intrastromal injections the presence of corneal edema, corneal epithelial erosion, or corneal neovascularization [13].

The management of fungal keratitis had been standardized using the Topical Systemic and Target Therapy (TST) protocol [14]. The first line of therapy was natamycin 5% concentration (50 mg/mL) hourly for the first 48 h then every 2 h until ulcer resolution. Subsequently, the dose was reduced to four times daily for another 3 weeks. If there was no improvement in the next 7–10 days, topical voriconazole 1% concentration (10 mg/mL) was added, at the same dosage as before. If there was still a poor response after 7–10 days of treatment, then intrastromal and/or intracameral injections of antifungals were performed and repeated 72 h apart [14]. All of them continued topical antifungal therapy during the cycle of intrastromal treatment. In this case series, patients non-responsive to topical therapy were treated with cycles of intrastromal injections of voriconazole (50 µg/0.1 mL) and amphotericin B (2.5 µg/0.1 mL).

TPK was reserved for recalcitrant cases not responding to targeted therapy or in cases of severe corneal thinning that contraindicated intrastromal injection or corneal perforation.

## 3. Results

Twenty patients were included in this case series, fourteen males and six females; the mean age was 57.05 ± 13.82 years. The mean presentation time at our institution was 33.8 ± 7.25 days from the onset of the symptoms. The average time between the culture test results and the beginning of the intrastromal injections was 15.0 ± 1.88 days. The mean duration of follow-up was 53.30 ± 11.27 days. The size of the infiltrate varied from 1.5 to 6.5 mm while the size of the epithelial defect varied from 1.0 to 6.2 mm. Twelve patients presented with a fixed hypopyon of less than 5 mm. The mean +/− number of intrastromal injections for each patient was 3.6 ± 1.0.

We identified the following risk factors for the development of fungal keratitis: traumatic contact with vegetables or soil (*n* = 14), previous corneal refractive surgery (LASIK, laser in situ keratomileusis, *n* = 1), topical steroid therapy for previous clinical conditions (*n* = 3), and neurotrophic keratitis (*n* = 2). 

Corneal scraping was performed in all patients after at least 48 h of washout from topical antibiotics and antimycotics. The organisms isolated were the *Fusarium* species (*n* = 5), *Aspergillus* species (*n* = 4), *Candida* species (*n* = 4), *Rhodotorula* species (*n* = 2), *Penicillium* species (*n* = 2), *Alternaria* species (*n* = 1), *Bipolaris* species (*n* = 1), and *Curvularia* species (*n* = 1) (Table 1).

At the time of presentation, the BCVA in all patients was less than 20/400, which improved after treatment to more than 20/400 in eleven patients; the BCVA improved to 20/80 in only two eyes. Only three patients had no improvement from the initial BCVA to the final BCVA due to central corneal leucoma.

All patients complained of ocular pain immediately after the injections but only one patient had to postpone the following cycle of injections due to acute ocular pain. One patient developed intrastromal and intracameral hemorrhages, which resolved after two weeks without any intervention.

In 16 patients, constituting 80% of the total, we achieved complete healing, defined as the resolution of the infiltrate and total re-epithelization (Figure 1).

Seven of the twenty patients required surgical intervention. In four patients, corresponding to 20%, we performed a penetrating keratoplasty (PK) à chaud one month from the first intrastromal injection since the infection showed no improvement (Figure 2). In the other three patients, three months after the last injection, we performed a PK to remove the remnant leucoma from the infection; patients who underwent penetrating keratoplasty gained a BCVA of at least 20/40 at 6 months follow-up. In all cases, no recurrence of mycotic keratitis was observed six months after the last injection.

## 4. Discussion

The management of fungal keratitis is challenging due to its insidious onset, non-specific clinical manifestation, and often ineffective therapies. Most topical antifungal agents have only fungistatic effects and are characterized by poor penetration into deep stroma. Therefore, the risks of recurrence, delayed ulcer healing, and corneal perforation are much higher than in bacterial keratitis [15,16,17]. Although voriconazole exhibits fungicidal properties and higher permeability than other antifungal agents, it still has poor penetration into deeper layers, resulting in a suboptimal therapeutic drug concentration at the site of infection [18,19,20]. Thus, intrastromal injections have been introduced to achieve adequate intrastromal penetration of antifungal agents. Several studies have reported outcomes of targeted therapy with voriconazole alone or combined with amphotericin B. Prakash et al. [11] reported the successful use of intrastromal voriconazole in three patients with deep non-healing fungal ulcers. Similarly, Tu et al. [21] reported favorable outcomes in *Alternaria* keratitis and Jain et al. [22] showed the success of intrastromal voriconazole in a fungal infection of the phacoemulsification site tunnel. Additionally, Fontana et al. reported using intrastromal injections to treat fungal interface keratitis after endothelial keratoplasty [23]. The role of intrastromal voriconazole becomes particularly crucial in treating recalcitrant mycotic keratitis. Indeed, infections poorly responsive to topical and systemic therapy often benefit from intrastromal injections. Sharma et al. [24] reported a more than 80% success rate in a prospective study of twelve eyes with fungal keratitis unresponsive to topical antifungals. In other studies, the success rate ranged from 70% [25] to 72% [7]. In our patients, we performed combined intrastromal injections of voriconazole (50 µg/0.1 mL) and amphotericin B (2.5 µg/0.1 mL) in eyes unresponsive to topical therapy with natamycin 5% and voriconazole 1%. None of our patients developed any toxic effects after injection. In fact, several experimental studies on rabbits’ corneas have shown safety in utilizing the intrastromal injection of antifungal substances at even higher concentrations than those used for our patients. Park et al. [13] showed that the intrastromal injection of voriconazole at a concentration higher than 100 µg/0.1 mL is toxic against endothelial cells and suggested a concentration lower than 50 µg/0.1 mL. Conversely, Qu et al. [26] demonstrated that the intrastromal injection of amphotericin B up to 5 or 10 µg/0.1 mL does not show toxicity for endothelial cells and corneal keratocytes.

We performed repeated injections in all patients. Because of the lack of knowledge of the pharmacokinetics of amphotericin B and voriconazole injected in the stroma, it is challenging to standardize the criteria for repeating intrastromal injections. Hence, the number of injections and the intervals between them must be determined based on clinical findings and may differ from patient to patient. The need for repeated injections to eradicate fungal infection has been reported in the literature. Sharma et al. [24] obtained good responses by performing two or more injections in ten out of twelve eyes treated. Likewise, Konar et al. [25] reported a rate of repeated injections of 75%. Considering the severity of recalcitrant infections treated with an intrastromal injection, in our study and the literature, it is clear that managing deep mycotic infections usually requires more than a single injection. 

In our observation, the infiltrate size and the height of the hypopyon were the main risk factors for poor treatment outcomes. These findings are consistent with those reported in the literature. Lalitha et al. [27] reported that ulcers of more than 14 mm^2^, hypopyons, and cultures positive for *Aspergillus* are risk factors for treatment failure. Similarly, Kalaiselvi et al. [7] and Konar et al. [25] have shown that infiltrated areas and the height of hypopyons significantly affect the outcomes. After intrastromal injections, we observed a reduction in ulcer, infiltrate, and hypopyon size in all patients.

Therapeutic penetrating keratoplasty is often the last option for treating recalcitrant fungal keratitis that is unresponsive even to targeted therapy [9]. However, the risk of recurrence is much greater in fungal keratitis than in bacterial; TPK is reported to eradicate the infection in 90% to 100% of bacterial keratitis cases but only in 69% to 90% of fungal ones, probably due to deeper penetration of fungal hyphae and poor response to antifungal drugs [12]. Since the recurrence of mycotic infection is the main cause of transplant failure in most of the reports, minimizing the risk of recurrence is crucial in their management. Better outcomes may be achieved with early intervention before perforation or scleral involvement [28]. As suggested in the literature, Xie et al. [29] showed that, after early TPK, 84.6% of grafts remained clear during follow-up. Similarly, Jain et al. [30], in a retrospective study of 28 patients, reported a rate of success of 80% for early TPK. However, the retro-iris exudates, coexisting endophthalmitis, degree of inflammation, and larger grafts affect the recurrence rate of the infection and graft survival [8]. Therefore, early keratoplasty is not always advisable, particularly in the most severe cases. In our case series, seven out of twenty patients required surgical intervention. In four of them, performing a PK à chaud was necessary. In these patients, repeating intrastromal injections was useful in reducing inflammation and ulcer size. Limiting the size of the infection allowed a smaller graft size to eradicate the infection, potentially improving further the keratoplasty outcomes. Indeed, graft size is one of the main risk factors for graft failure in therapeutic penetrating keratoplasty [8]. Chatterjee et al. outlined the ability of fungi to infiltrate the deep corneal stroma as a major risk factor for the recurrence of fungal keratitis [12]. Therefore, to improve the safety of the procedure in terms of preventing recurrence, after suture placement, we performed a circular intrastromal injection in the host stroma [31].

## 5. Conclusions

In summary, the combined intrastromal injection of voriconazole and amphotericin B is an effective therapy for managing recalcitrant deep mycotic infections. Compared to topical antifungal therapy, the main advantage is the higher drug concentration achieved in the deep corneal stroma. Furthermore, an intrastromal injection before PK à chaud is worthwhile in reducing ulcer size and, thus, the graft diameter, preventing the recurrence of infection.

## Figures and Tables

**Figure 1 microorganisms-12-00922-f001:**
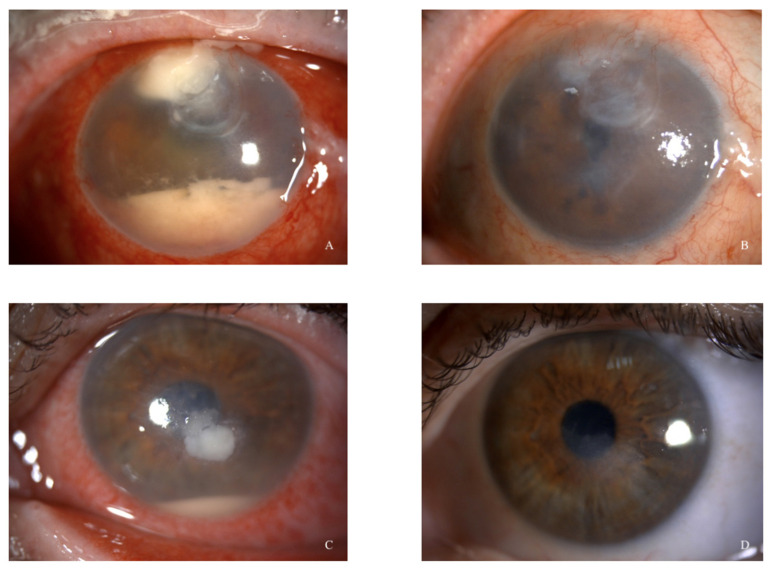
Cases 1 (**A**) and 5 (**C**). Slit lamp anterior segment photographs at presentation. (**B**,**D**) Anterior segment photographs three months after intrastromal injections.

**Figure 2 microorganisms-12-00922-f002:**

Case 3. (**A**) Slit lamp anterior segment photographs at presentation. (**B**) One month after intrastromal injections. (**C**) Six months after penetrating keratoplasty ‘a chaud’.

**Table 1 microorganisms-12-00922-t001:** Presentation and outcome of patients with recalcitrant fungal keratitis that received intrastromal injections of voriconazole (50 µg/0.1 mL) and amphotericin B (2.5 µg/0.1 mL).

Pt	Days of Presentation	Days before INJ	Initial BCVA	Final BCVA	Size of Infiltrate	Size of Epithelial Defect	N° of Cycles	FU (Days)	Aetiology
1	25	15	LP	HM	3.5 × 2.8 mm	5.2 × 4.4 mm	5	40	*Rhodotorula*
2	32	17	CF	20/100	2.5 × 2.2 mm	2.5 × 1.5 mm	2	65	*Fusarium*
3	42	14	LP	LP	5.5 × 4.3 mm	2.8 × 2.5 mm	3	32	*Alternaria*
4	28	18	HM	20/200	4.0 × 3.5 mm	1.5 × 1.0mm	6	75	*Bipolaris*
5	40	16	20/800	20/80	3.0 × 2.4 mm	2.2 × 1.8 mm	4	55	*Fusarium*
6	28	19	LP	HM	5.0 × 4.5 mm	2.5 × 1.5 mm	2	58	*Aspergillus*
7	32	15	CF	20/800	2.8 × 2.3 mm	3.2 × 2.6 mm	5	65	*Fusarium*
8	40	14	20/800	20/200	3.5 × 2.5 mm	6.2 × 4.2 mm	3	62	*Curvularia*
9	38	18	CF	20/100	4.2 × 2.3 mm	3.0 × 1.8 mm	4	40	*Candida*
10	25	20	HM	CF	4.8 × 3.5 mm	3.2 × 1.5 mm	3	52	*Aspergillus*
11	37	18	20/400	20/200	2.5 × 2.2 mm	4.2 × 3.8 mm	5	46	*Candida*
12	42	17	CF	20/800	4.2 × 3.7 mm	4.5 × 4.2 mm	4	56	*Fusarium*
13	31	20	HM	HM	6.5 × 4.3 mm	2.5 × 2.2 mm	3	43	*Penicillium*
14	48	16	CF	20/200	4.8 × 3.8 mm	1.8 × 1.2mm	3	55	*Candida*
15	40	17	20/400	20/80	1.8 × 1.5 mm	2.5 × 1.4 mm	2	74	*Penicillium*
16	24	19	HM	HM	5.0 × 4.8 mm	2.6 × 1.9 mm	5	48	*Aspergillus*
17	29	15	20/400	20/100	2.8 × 2.5 mm	3.5 × 3.1 mm	3	51	*Fusarium*
18	41	18	20/800	20/200	3.5 × 3.2 mm	6.2 × 4.4 mm	4	53	*Rhodotorula*
19	33	14	CF	20/100	3.6 × 2.5 mm	3.2 × 2.5 mm	4	58	*Candida*
20	21	16	LP	CF	4.8 × 4.6 mm	2.8 × 2.4 mm	3	38	*Aspergillus*

Pt: patient; INJ: injection; BCVA: best corrected visual acuity; FU: follow-up; LP: light perception; CF: count finger; HM: hand motion.

## Data Availability

Data are contained within the article.

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
