# Peer review of "Efficacy of the Combined Intrastromal Injection of Voriconazole and Amphotericin B in Recalcitrant Fungal Keratitis"

_microorganisms, 2024, doi:10.3390/microorganisms12050922_

Round 1

Reviewer 1 Report

Comments and Suggestions for Authors

The authors explore the effectiveness of intrastromal injections of voriconazole and amphotericin B for managing refractory fungal keratitis. While certain severe cases still necessitate corneal transplantation, the study's outcomes demonstrate the combined treatment's efficacy. However, a question arises regarding the authors' criteria for defining "poorly responding" when selecting patients for the study. Additionally, although the authors mention monitoring the toxicity associated with stromal injections, they don't provide specific information on how toxicity was assessed or its outcomes.

Reviewer 2 Report

Comments and Suggestions for Authors

The authors present a manuscript describing the efficacy of intrastromal antifungal treatment of recalcitrant fungal keratitis. Several revisions should be made to strengthen the manuscript. 

Were antifungal susceptibility tests performed for the fungal isolates to natamycin, voriconazole, and amphotericin B? If so, this data should be included. If not, then state the reason. 

Define BCVA in the abstract. Microorganisms is not an ophthalmic journal therefore all common ophthalmic abbreviations must be defined. 

Use the conventional abbreviation for micrograms (µg) rather than µgm. 

Use the standard American numerical system for decimals, for example, 2.5 rather than the European numerical system (2,5). This must be consistent throughout the manuscript. 

Italicize the genus of all fungal species throughout the manuscript and Table 1. 

Capitalize Figure 1 and Figure 2 when cited in the text. 

Define all abbreviations in Table 1 (both headers and data). In addition, include the means and standard deviations of the data from all columns when applicable. 

Figure 1 Legend: Revise “Case 1-5. (A-C)” to “Cases 1 (A) and 5 (C).” Also revise “(B-D)” to “(B, D)”.

Line 52, revise the term “therapeutic keratoplasty (TPK)” to “therapeutic penetrating keratoplasty (TPK)”.

Line 74, capitalize Gram.
